# Cerebro-spinal flow pattern in the cervical subarachnoid space of healthy volunteers: Influence of the spinal cord morphology

Lugdivine Leblond[1,2], Patrice Sudres[1,2], Morgane Evin[1,2]*

1 Laboratoire de Biomécanique Appliquée, UMRT24, Aix Marseille Université, Marseille, France, 2 iLab-Spine - Laboratoire International en Imagerie et Biomécanique du Rachis, Marseille, France

* morgane.evin@univ-eiffel.fr

**Data Availability Statement:** All relevant data are within the manuscript and its Supporting information files.

## Abstract

### Introduction

Toward further cerebro-spinal flow quantification in clinical practice, this study aims at assessing the variations in the cerebro spinal fluid flow pattern associated with change in the morphology of the subarachnoid space of the cervical canal of healthy humans by developing a computational fluid dynamics model.

### Methods

3D T2-space MRI sequence images of the cervical spine were used to segment 11 cervical subarachnoid space. Model validation (time-step, mesh size, size and number of boundary layers, influences of parted inflow and inflow continuous velocity) was performed a 40-year-old patient-specific model. Simulations were performed using computational fluid dynamics approach simulating transient flow (Sparlart-Almaras turbulence model) with a mesh size of 0.6, 6 boundary layers of 0.05 mm, a time step of 20 ms simulated on 15 cycles. Distributions of components velocity and WSS were respectively analyzed within the subarachnoid space (intervertebral et intravertebral levels) and on dura and pia maters.

### Results

Mean values cerebro spinal fluid velocity in specific local slices of the canal range between 0.07 and 0.17 m.s⁻¹ and 0.1 and 0.3 m.s⁻¹ for maximum values. Maximum wall shear stress values vary between 0.1 and 0.5 Pa with higher value at the middle of the cervical spine on pia mater and at the lower part of the cervical spine on dura mater. Intra and inter-individual variations of the wall shear stress were highlighted significant correlation gwith compression ratio (r = 0.76), occupation ratio and cross section area of the spinal cord.

**Funding:** The authors received no specific funding for this work.

**Competing interests:** The authors have declared that no competing interests exist.

## Conclusion

The inter-individual variability in term of subarachnoid canal morphology and spinal cord position influence the cerebro-spinal flow pattern, highlighting the significance of canal morphology investigation before surgery.

## Introduction

The cerebrospinal fluid (CSF) flows within the subarachnoid canal in the dural sac and its major part is produced by the choroid plexus in the highest ventricles in the brain. It is redirected via the third ventricle to end up in the fourth ventricle and flows through the subarachnoid space (SS) of the brain and spinal cord. It is then reabsorbed into the dural venous sinuses by the arachnoid granulations [1].

The CSF flows in the subarachnoid canal with a pulsation set on the cardiac cycle. The phase shift between the CSF signal and the arterial signal has been shown as age and gender dependent [2]. This pulsation is patient-dependent as depending of the variation of the cerebral volume, the spinal vascularization but also of the anatomical and hemodynamic criteria. Additionally, the CSF flow alterations taking part in the physio-pathological mechanisms can be measured through velocities and pressures within the subarachnoid canal [3,4]. The CSF role could then be hypothesized in degenerative and congenital pathologies such as Chiari, syringomyelia or myelopathy [5,6]. CSF action has been particularly highlighted in syringomyelia in which a fluid cavity creates inside the spinal cord and is promoted by CSF flow. Syringomyelia main aetiologia is congenital Chiari pathology or traumatic [5] for which CSF measurement is part of the diagnostic.

The geometry of the canal is influenced by age and gender, being either obstructed because of lesions, pathologies or normal aging such as osteophytes. The pulsation of the CSF is then modified physiologically [7,8]. The cervical spine morphology has been greatly characterized in the literature, reporting inter-individual variation in term of spinal cord positioning with the SS as well as spinal cord morphology itself [9–11]. The relationship between CSF flow pulsation and morphology is still to be investigated and could enable a clear investigation of the impact of inter-individual morphological variations.

Simplified cylindrical models of the SS have enabled development of equation based modelling questioning material properties of the dural sac as well as location of the spinal cord within the SS [12].

Additionally, CSF simulations with fluid-structure interaction have also been described [13]. Numerical models can be conducted using simplified geometry of the canal [14,15] or using individual-specific 3D canal, offering better anatomical details [16,17]. More and more studies use a pulsatile velocity profile either from a literature defined profile or a PC-MRI defined profile [3]. Partial validation of the developed models has been proposed [4,18].

Regarding quantification of CSF flow, the Wall Shear Stress (WSS), representing the effects of shear flow on the wall, Oscillatory Shear Index (OSI), representing the WSS vector re orientation from a main flow, or local Reynolds number can be derived from simulation. The local Reynolds number was especially used to describe and detect possible turbulence resulting from CSF pulsation and SCS morphology complexity [8]. Lack of complicated geometries modelling is underlined in different study such as in [19] in which differential intracranial pressure in hydrocephalus patients were investigated to assess the contribution of cardiac and respiratory effect on CSF.

The aim of this work is three folds: 1) to simulate CSF cervical flow in the subarachnoid space using CFD in 11 healthy volunteers after model validation, 2) to quantify inter individual variability of the CSF flow pattern, 3) to analyze the influence of geometry on the CSF subarachnoidal flow.

## Methods

The overall approach was to build a subject-specific numerical model of the CSF flow in the subarachnoid space using a Computational Fluid Dynamics (CFD) approach while performing a sensitivity analysis to assess the simulated flow.

### Solver

In this study, the numerical simulations were performed using the CFD solver Altair AcuConsole® (version 2019.1, Altair Engineering. Inc., Troy. Michigan, United States). The CSF flow was assumed to be transient using the Sparlart-Almaras turbulence model considering complexity of the geometry of the canal and previous observation from the literature [4] and despite the following fluid indices: Reynolds number (187 to 352) of laminar flow, Strouhal number of 2.33 (considering 70 bpm and mean velocity of 0.03 m.s$^{-1}$ and amplitude of 0.06 m) and Womersley number in individuals (6.2–7.3) [20]. The fluid was assumed to have the same physical properties as the water at body temperature (37°C) i.e. fluid density $\rho$ = 1000 kg.m$^{-3}$ and kinematic viscosity $\upsilon$ = 0.7x10$^{-6}$ m$^2$.s. An initial pressure of 14 mmHg was also imposed [20].

### MRI acquisition of healthy volunteers and morphological characterization

3D anatomical cervical canals (from C2 to C7) 3D T2-space sequence MRI images (isotropic voxel size 1 mm$^3$) were acquired in neutral position for 11 healthy volunteers (5 females and 6 males, mean age: 29.09 ± 8.42y., BMI: 23.64 ± 2.44 kg/m$^2$—Table 1. The study protocol was approved by the local Ethics Committee (2011-A00929-32) and all subjects signed the

**Table 1. Individual demographic and morphological characteristics.** CSA for Cross Section Area, Ecc for eccentricity, CR and OR respectively compression and occupational ratios.

| Population | 11 | | |
|---|---|---|---|
| Gender (% female) | 45.45% | | |
| Weight (kg) | 74.64±12.27 | | |
| Size (cm) | 177.18±9.67 | | |
| Age (y.) | 29.09±8.42 | | |
| BMI (kg.m2) | 23.64±2.44 | | |
| Length (mm) | 122.13±7.59 | | |
| LAC (mm) | 118.01±5.21 | | |
| LPC (mm) | 107.6±5.57 | | |
| LSC (mm) | 141.56±6.77 | | |
| | **Mean Spine** | **Vertebrae C3** | **Vertebrae C5** | **Vertebrae C7** |
| CSA_SC (mm2) | 80.02±7.8 | 80.22±9.54 | 83.46±10.47 | 65.16±6.71 |
| CSA_CSF (mm2) | 172.2±43.49 | 131.9±33.67 | 136.18±27.75 | 146.6±32.78 |
| OR | 0.34±0.04 | 0.38±0.04 | 0.38±0.03 | 0.31±0.04 |
| Ecc_AP (-) | -6.89±10.25 | -20.44±13.77 | -0.97±16.57 | -23.01±21.82 |
| Ecc_LR (-) | 3.86±4.83 | 5.01±6.52 | 1.31±9.19 | 3.14±12.05 |
| CR | 0.63±0.03 | 0.62±0.04 | 0.57±0.05 | 0.6±0.03 |

informed consents from January 2017 to December 2018. The 3D models analyzed in this study are derived from a previously published study [9]. The cervical subarachnoid spaces were defined using a semi-automatically segmentation tool, ITK-SNAP. Subarachnoid canal morphology of the healthy volunteers of such study has been characterized in [9]. Briefly, the lengths of the cervical canal are defined for anterior column (AC), posterior column (PC) and spinal cord (SC); Cross Section Area (CSA) are defined for spinal cord (SC) and CSF, Eccentricity (Ecc) are defined for antero-posterior (AP) and left-right (LR). Compression (CR) and Occupational (OR) ratios are defined respectively the AP diameter on transverse diameter of the spinal cord and CSA of spinal cord on CSA of the canal.

## Patient-specific subarachnoid space model and computational mesh

The model geometries were imported in Altair Hypermesh® (version 2019.1. Altair Engineering. Inc., Troy. Michigan, United States). Tubes of 10 cm were added by extending the upper and lower surfaces [8] to mitigate boundary effects. The skewness measurements were performed according to AcuConsole Training Manual with the ration ideally tending to 0 for tetrahedrons.

Mesh refinement was performed near the walls grid quality using y+ value (non-dimensional distance to the wall) computation as (Eq 1):

$$y+ = \frac{yu_\tau}{v} \tag{1}$$

Where y is the distance from the wall, or the first cell height from the boundary (in m), $v$ is the kinematic viscosity (in m.s-2) and $u_\tau$ is the frictional velocity (Eq 2).

$$u_\tau = \sqrt{\frac{\tau_\omega}{\rho}} \tag{2}$$

With $\tau_\omega$ is the WSS (in Pa) and $\rho$ is the density (in kg.m³).

The value of the first layer thickness was 0.1 mm and the boundary layer growth rate was 1.2.

## Boundary conditions

The CSF pulsation modeled by an average velocity profile on the entry plane y similar to reported physiological CSF flow pattern [7]. By simulating the pulsation on the upper surface of the tube, boundary effects linked with the inflow boundary are minimized.

Measurements of the CSF velocity by 4D PC-MRI tend to prove that the velocity is not equally distributed on the surface of the canal [4]. A finer velocity profile was tested by dividing the upper surface of the geometry into four parts of equal areas and applying a percentage of the mean value velocity profile derived from 4D PC-MRI measurements along the cervical spine (Fig 2C). A zero-pressure condition was applied on the lower boundary.

The outer and inner boundaries of the subarachnoid space ware considered as rigid and a no-slip boundary condition was applied at the pia and dura matter walls as in [21].

## Model validation

Model validation and sensitivity analysis as defined by the Food and Drug Administration [22] and also provided for CSF in [3,23] aims at providing proof of reliability of the results, referred as value of interest, and its independency to several parameters of the model setting. The sensitivity analysis was conducted on one healthy volunteer (42 y., M). Each model was tested with:

- 20 simulated CSF cycles to assess time-step and period independence [3]

- 5 element sizes: 0.4, 0.6, 0.8, 1 and 1.2 mm (Fig 1)

- 7 numbers of boundary layer: 4, 5, 6, 7, 8, 9 and 10

- 6 thicknesses of layer: 0.025, 0.05, 0.075, 0.1, 0.125 and 0.15 mm

- 3 time-steps: T/50 (20ms), T/100 (10ms) and T/200 (5ms), where T is the length of one CSF cycle (here 1s)

In the present study, the initial quantities of interest were the three components of velocity, components of the WSS and y+ value.

Criteria of validation of the model were defined as no variation or a variation lower than 10% (or 5%) compared from a reference.

## Post-processing and fluid indices computation

Region of interests were defined for 6 specific slices located at 6 different levels: 3 at intra-vertebral levels and 3 at the nerve's roots (Fig 2D–2E). Exportations of the results is performed

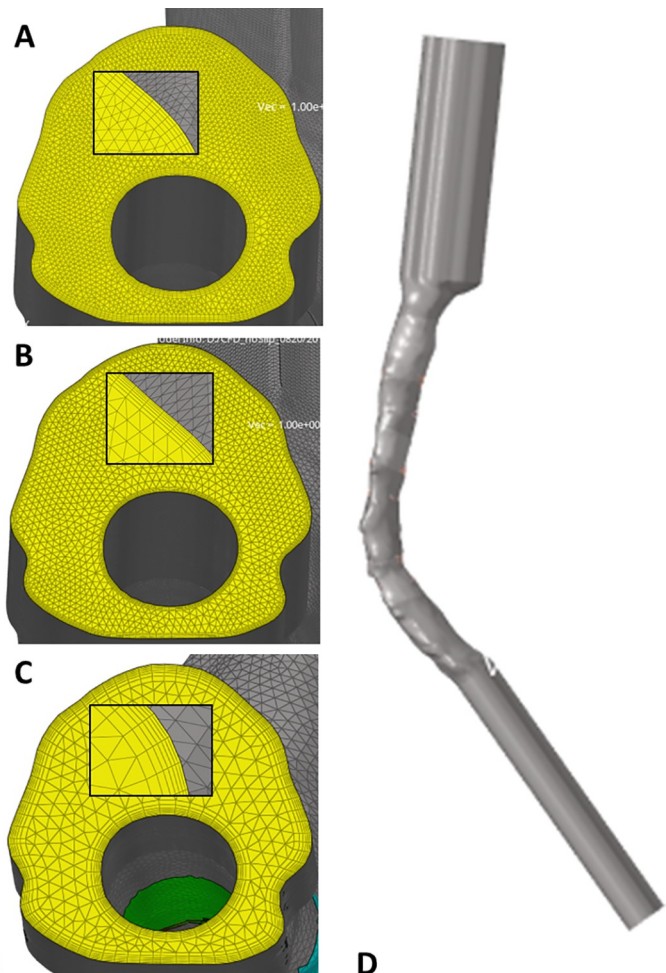

**Fig 1. Boundary conditions and post-processing settings.** Boundary condition definition (A) with boundary inflow with constant velocity addition of 0.005 and 0.01m/s or partial patterns (A and B, post: Posterior, ant: anterior, lat: Lateral), sectional boundary inflow definition (B) and percentage of the bulk flow (C). Post-processing part definition for WSS (D) and flow (E).

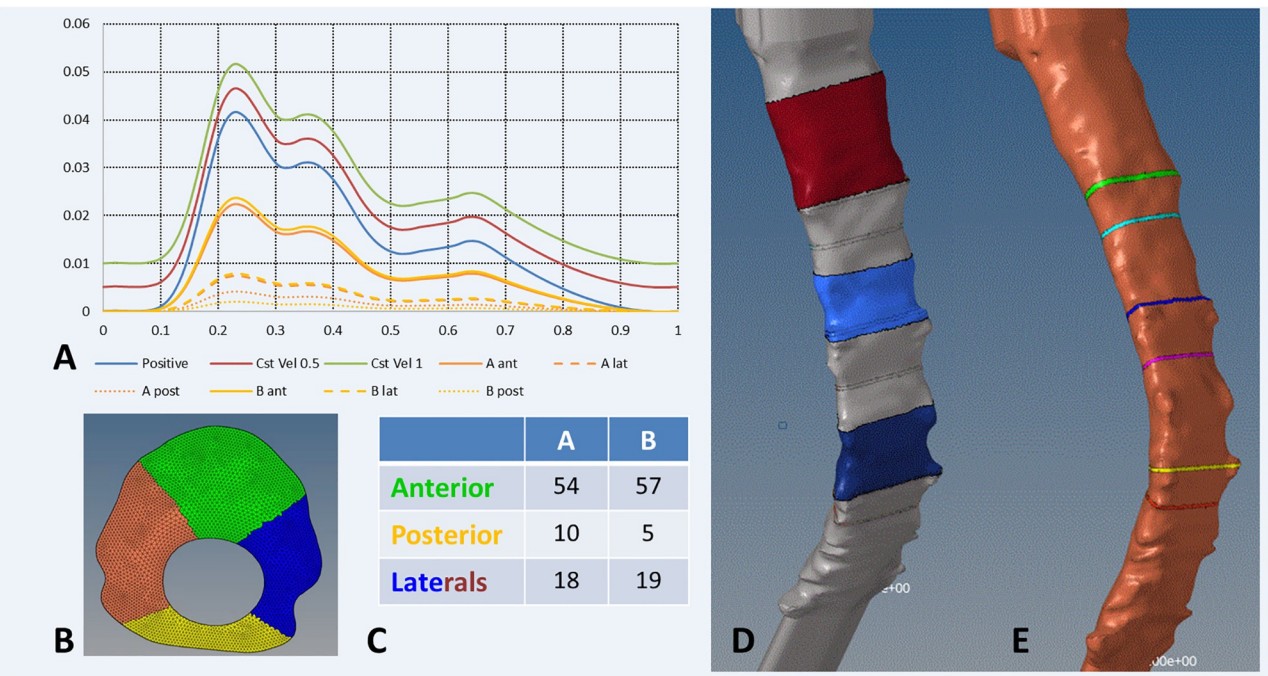

**Fig 2. Flow and WSS representation.** Mean velocities along the cervical canal (A), Max velocities along the cervical canal (B), mean WSS along the cervical canal (C) and max WSS along the cervical canal (D).

using Altair HyperView® (version 2019.1. Altair Engineering. Inc., Troy. Michigan, United States) at each node of the region of interests. Velocity components, WSS components, values of pressure and y+ value were extracted at different time-step.

The y+ value and WSS were directly assessed by the solver. WSS was determined as (Eq 3):

$$WSS = \mu \left( \frac{\partial U}{\partial y} \right)_{y=0} \tag{3}$$

Where µ is the dynamic viscosity, U is the flow velocity parallel to the wall, y is the distance to the wall.

## Statistics

Shapiro tests were used to determine if the data follow a normal distribution. T-test will be used for comparison between groups of normally distributed data and Wilcoxon otherwise. Significant threshold is defined as $p<0.05$. Influence of morphological parameters on fluid indices will be tested using MANOVA after one-to-one ANOVA test for inclusion in MANOVA.

## Results

### CSF flow pattern

CSF flow was quantified by computing maximum and mean values of velocity within the SS and WSS on pia and dura mater at different spine locations for all healthy volunteers (Table 2). Fig 3 shows that the flow is inhomogeneously distributed along the subarachnoid canal as the mean and maximum data are widely distributed at C5 Roots and C7 Intra. Maximum velocity

**Table 2. Velocities and WSS values averaged on the 11 individuals.**

| | Velocity (m/s) | | | | WSS (Pa) | | | |
| --- | --- | --- | --- | --- | --- | --- | --- | --- |
| | Intra | | Roots | | Dura | | Pia | |
| | mean | max | mean | max | mean | max | mean | max |
| C3/C2 | 0.13±0.04 | 0.21±0.07 | 0.09±0.03 | 0.15±0.04 | 0.08±0.004 | 0.18±0.08 | 0.09±0.04 | 0.16±0.07 |
| C5/C4 | 0.09±0.03 | 0.15±0.04 | 0.13±0.05 | 0.05±0.22 | 0.09±0.004 | 0.16±0.07 | 0.18±0.1 | 0.35±0.2 |
| C7/C6 | 0.13±0.05 | 0.22±0.09 | 0.11±0.03 | 0.03±0.18 | 0.18±0.01 | 0.35±0.2 | 0.19±0.1 | 0.25±0.12 |

values are found on C3/C2 and C7/C6 Intra and C3/C2 Roots. Mean values of velocity oscillate between 0.07 and 0.17 m.s$^{-1}$ and maximum values oscillate between 0.1 and 0.3 m.s$^{-1}$.

## WSS analysis

Discrepancies are also observed when studying mean and maximum values of WSS using medians comparisons (Table 2) which depict the complexity of the CSF pattern within the studied sections. Maximum WSS values vary between 0.1 and 0.5 Pa. When comparing WSS distributions through planes, regions with higher friction are identified at C4/C5 pia and C6/C7 dura reaching 0.35 dura mater (Fig 3). Additionally, when depicting the WSS at its peak value, higher values are located at the vertebral body in all canal and amplitude of the peak varies according with the individuals (Fig 4).

## Correlation between morphological indices and velocity/WSS

Correlations between mean and maximum values of velocity and WSS at different locations and several parameters such as age, dimensions of the subarachnoid canal and morphological indices are reported. Correlations were not significant as r<0.5 for all parameters except the Compression Ratio (CR) parameter (Table 3). Good correlations are observed between morphological indices and mean/maximum CSF flow at roots and between morphological indices and mean/maximum WSS values at Pia (r>0.74 and r>0.71 respectively- Fig 5).

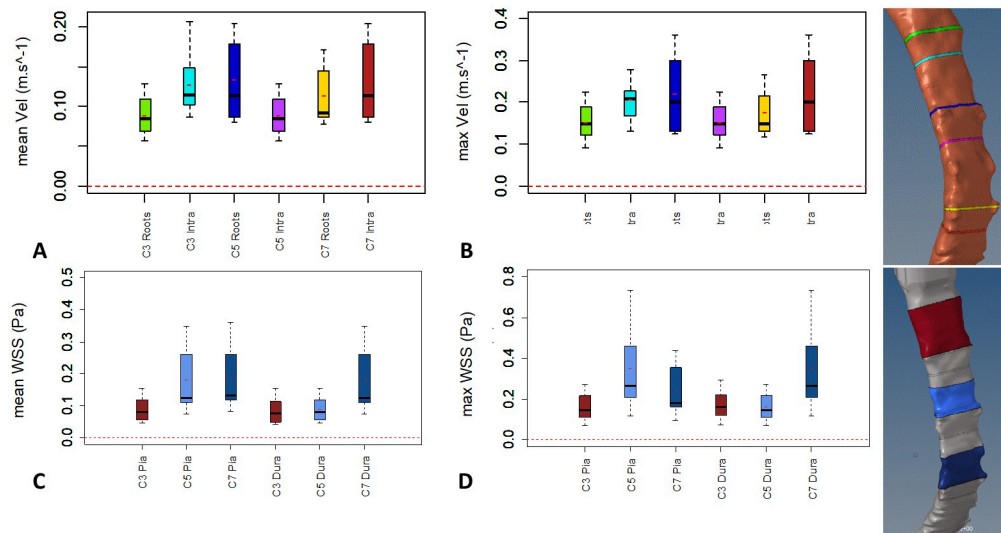

**Fig 3. Wall shear stress (Pa) depiction for four healthy volunteers: Maximum and minimum cross-sectional area CSA (of spinal cord SC and CSF) and compression ratio CR.**

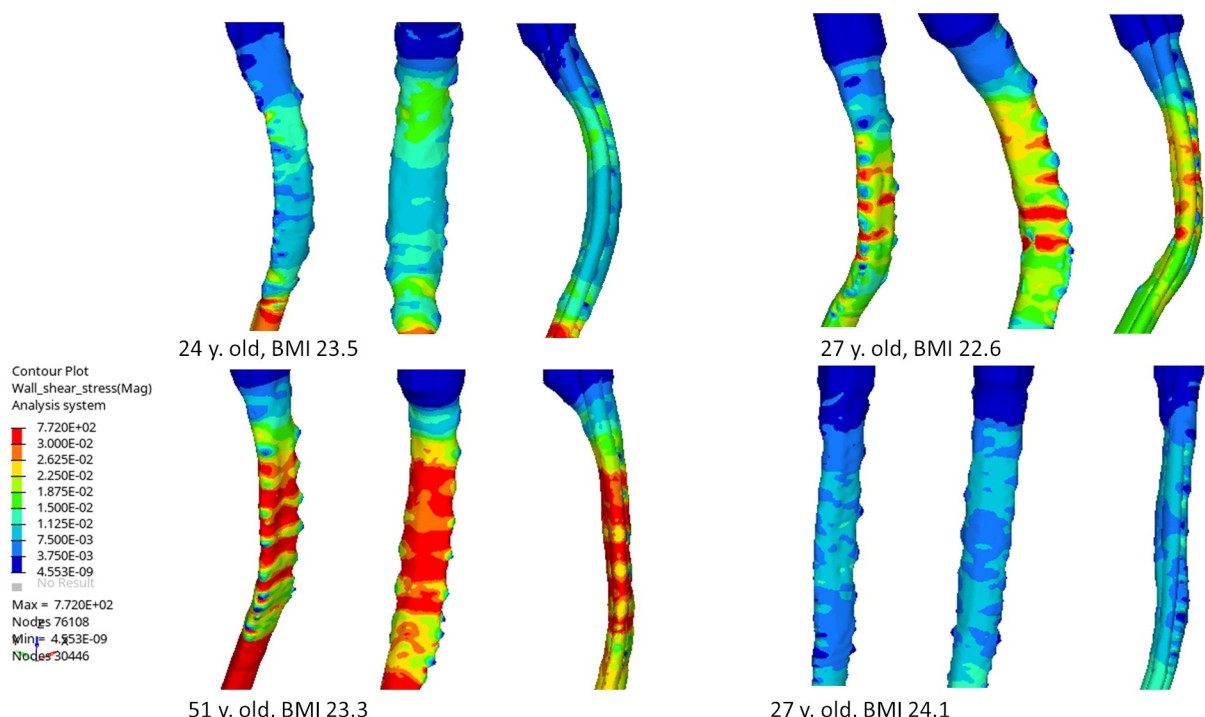

**Fig 4. Representation correlation indices morphological canal parameters and fluid indices.** Mean velocity value (A) and Max velocity values (B) at nerve roots according with Compression Ration CR; Mean WSS (C) and max WSS (D) at pia matter PM according with CR.

## Discussion

Improvement in the understanding of the CSF dynamics with quantification of inter-individual variability is the first step to better comprehend the pathophysiological mechanisms in order to assess and find treatment options for central nervous system (CNS) pathologies and spinal cord injuries. This study provided a detailed sensitivity analysis, which is necessary to assess the accuracy of the simulated flow as well as individual-specific numerical model on a

**Table 3. Correlation between morphological parameters and flow measurements.**

| r value | Velocity (m/s) | | | | WSS (Pa) | | | |
|---|---|---|---|---|---|---|---|---|
| | Intra | | Roots | | Dura | | Pia | |
| | Mean | Max | Mean | Max | Mean | Max | Mean | Max |
| Age (y.) | 0.44 | 0.39 | 0.48 | 0.46 | 0.39 | 0.37 | 0.43 | 0.43 |
| Length (mm) | 0.34 | 0.34 | 0.33 | 0.32 | 0.30 | | 0.32 | |
| LAC (mm) | | | | | | | | |
| LPC (mm) | | | 0.36 | 0.36 | | | | |
| LSC (mm) | | | | | | | | |
| CSA_SC (mm2) | 0.35 | 0.33 | | | 0.51 | 0.44 | 0.31 | |
| CSA_CSF (mm2) | 0.51 | 0.48 | 0.46 | 0.45 | 0.34 | 0.31 | 0.40 | 0.41 |
| OR | | | 0.34 | 0.39 | | | | 0.35 |
| Ecc_AP | | | 0.43 | 0.42 | | | 0.39 | 0.47 |
| Ecc_LR | | | | | | | | |
| CR | | 0.30 | 0.76 | 0.75 | 0.40 | | 0.74 | 0.73 |

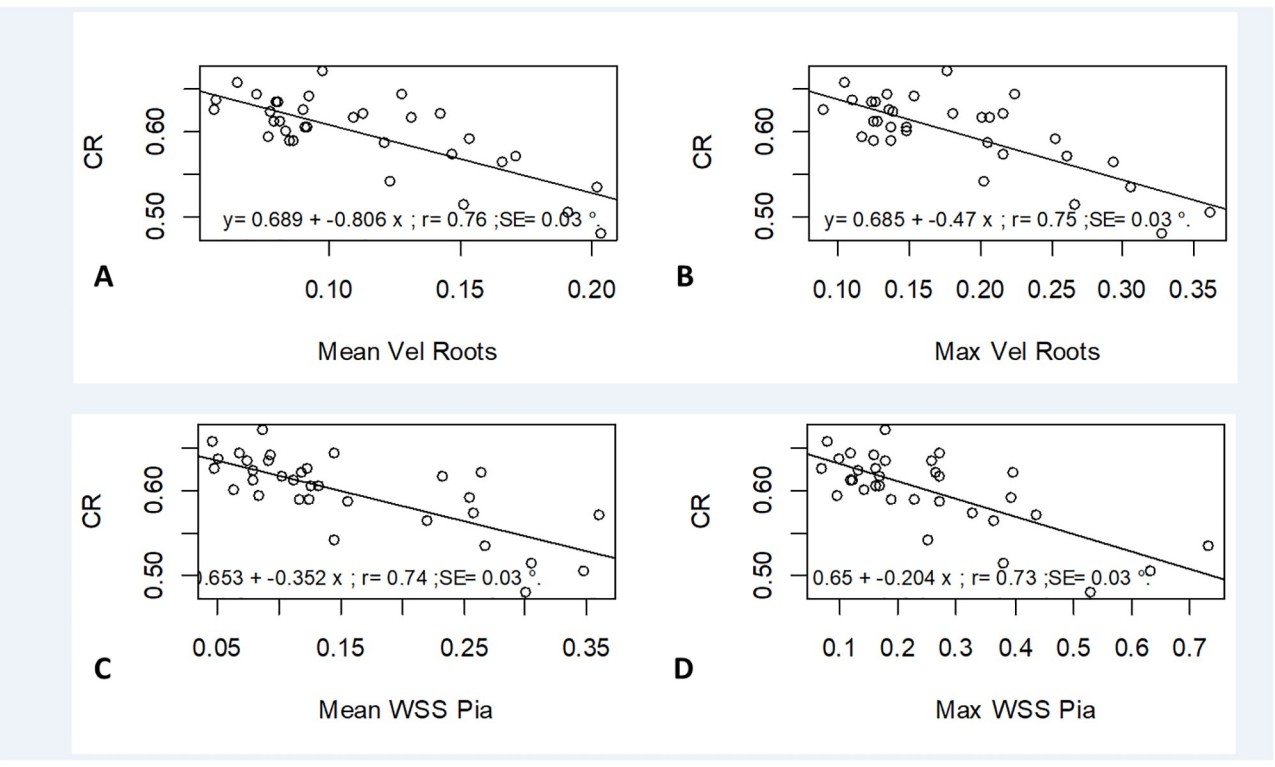

**Fig 5.**

population of eleven healthy volunteers. Based on this model validation, CSF flow were simulated within 11 healthy volunteers SS and correlations with morphology of the spinal cord within the SS were investigated.

## Sensitivity analysis

Final simulations have a global mesh size of 0.6 mm and 6 layers of 0.05 mm layers size running for 15 cycles and time step of 20 ms. While the use of a smaller mesh size (0.4) could refine the results, the 0.6 mesh size was furtherly chosen in order to lower the computation time.

According to the recommendations, y+ values inferior to 100 ensure the possibility to capture the viscous effects. y+ value for CSF flow simulation allows to understand the physics of the WSS close to the walls of the geometry [24,25]. Concerning the number of simulated cycles, the results were found to converge at the end of the tenth cycle. In this work, a higher number of CSF cycle simulated is chosen compared to the literature [23,26]. A time-step of 20 ms was chosen. Considering a higher time-step could lead to miss the viscous effects near the walls. Finally, the inflow modification with a constant velocity addition did not modified the WSS results and the influences of the optimized inflow with four section were smoothed by the tube addition.

## Flow over the cervical spine

Our findings demonstrated that the CSF flow is inhomogeneous along the cervical spinal. The section velocity results suggest that peaks are situated on C3/C2 and C7/C6 Intra and C5/C4

Roots. Comparison with the literature could be performed either with the MRI 4D-flow measurements [17,27] or when comparing with other numerical simulation work.

Higher peak velocities were found in our study (between 10 and 30cm/s) than in 4D-MRI and other simulation (mean velocities between 7 and 17cm/s). Such discrepancies could be due to constant velocity addition as well as resulting from averaging of the 4D-flow acquisition on the time-step. Additionally, the quantification of the velocity as mean and average values has been shown to be necessary as fully describing the flow pattern. Such overestimation could be due to depiction in the literature of averaged value only as well as underestimation resulting from flow measurements.

## WSS and physio-pathological mechanisms

Maximum WSS values range from 0.09 Pa to 0.14 Pa (mean 0.14±0.04 Pa) on dural walls and from 0.13 to 0.2 Pa (mean 0.16±0.03 Pa) on pia mater. Such values are comparable to the one reported in the literature. Indeed, in [14,23], values of peak WSS between 0 and 1 Pa were measured. Such values are relatively low and could result from small discrepancies from non-individual-specific boundary condition.

Our results showed higher value of WSS between the discs at the level of vertebral bodies and lower WSS at the nerves roots location (Fig 4). Inter-individual differences could be noticed and relation with age could be studied through morphological influences on the CSF flow.

## Correlation between morphological indices and velocity and WSS

Our results stipulate that there is a significant correlation between compression ratio of the spinal cord and WSS measurements and that such correlation is higher for compression ratio than occupation ratio. While relationship between occupation ratio and flow is expected as reducing the section increases the fluid velocity, higher correlation with compression ratio shows that the geometry of the spinal cord is and its changes among individuals should be carefully studied. With spinal cord antero-posterior diameter increasing and transverse diameter of the spinal cord decreasing, the spinal cord could be considered flatter letting more space in the spinal canal. The WSS and velocity at intra vertebral level and nerve root level are found lower. Those results highlight the influence of the spinal cord shape within the SS rather than focusing on the occupation of the SC within the canal. This is also depicted by the WSS pattern (Fig 4) when comparing 1) low CSA (24y. old individual, BMI of 23.5 kg.m$^{-2}$) and high CSA (27y. old individual, BMI of 22.6 kg.m$^{-2}$) and 2) low CR (51 y. old individual, BMI of 24.1 kg.m$^{-2}$) and high CR (27y. old individual, BMI of 24.1 kg.m$^{-2}$). Additionally, curvature of the spinal cord could have been furtherly studied. In perspective, the attachment of the spinal cord at the lower brain level as well as along the while spine with denticulate ligament and nerve roots and the influence of such attachments could be questioned.

## Clinical application

This study is the first to fully address and report correlation between morphology changes and CSF flow pattern. It is a step forward to the understanding of the fluid mechanics and biomechanics of the CSF interacting with spinal cord especially in impact and in post-traumatic context. In healthy volunteers, it shows that the patient-specificity should be addressed in term of morphology before further addressing change on material properties of the pia and dura mater and fluid-structure interaction. Such knowledge on CSF flow pattern could also help towards quantification of the CSF using phase contract MRI in 2D [28,29] or 4D [30] for diagnosis CSF flow alteration. This study in healthy volunteers could then be completed with further work in pathological context such as congenital (Chiari malformation for example) and

post-traumatic conditions as well as extending to cranio-cervical junction. Numerical models could also be used for predicting surgical outcomes and serve as decision support system.

## Limitations

The presence of anatomical structure like the arachnoid trabeculae, blood vessels or cord nerve roots and denticulate ligaments was not considered and has been shown to be relevant [23,31].

Furthermore, the addition of compliance of the canal components could lead to significant differences in the observation of the CSF velocity. Elastic properties of neural tissues need to be added to the model to represent the tissue compliance. Indeed, the influences of interactions between the CSF and the surrounding tissues has been investigated [32]. Initialization of the velocity could have been considered to limit the simulation time.

The lack of neural tissue motion can also be noted as this study was rigid walled. Several studies quantified the bulk motion of neural tissues. Bunck et al. [27] and Yiallourou et al. [17] proved that the tonsils and spinal cord motions result in the increasing of the CSF velocity in regions near the foramen magnum in pathological context. CFD simulation supplemented with moving boundary on the upper cervical spine or a fluid-structure interaction model might lead to more realistic results [23,33].

## Conclusion

This work finally provides some insight of the relationship between CSF flow and morphology of the canal while providing values on a larger number of individuals as compared to the previously published literature. The spinal cord shape within the canal is then highlighted as correlating with velocity and WSS of the CSF within the SS. This study highlights the necessity to include patient-specificity in the simulations of the CSF flow. Further investigation will be required to understand the associated physiology of the CSF pulse in the perspective of describing it in the context of altered meningeal tissue.

## Supporting information

**S1 Fig. Sensitivity analysis settings.** Segmentation, mesh, mesh refinement, layer mesh size (1.2–A, 1 -B, 0.6-C), pipes addition (D).
(PNG)

**S2 Fig. Sensitivity analysis on boundary layers number.** Velocity and WSS profile at C7 and C6 locations with boundary layers number varying from 4 to 10 layers of 0.05mm.
(PNG)

**S3 Fig. Size of the boundary layers.** Velocity and WSS profile at C7 and C6 locations with the size of boundary layers varying from 0.025 to 0.15mm.
(PNG)

**S4 Fig. Sensitivity analysis of the number of simulated cycles.** Velocity and WSS profile at different locations for 20 cycles.
(PNG)

**S5 Fig. Sensitivity analysis on the inflow profile—Velocity and WSS profile at C7 and C6 locations with inflow profiles as described in Fig 2.**
(PNG)

**S1 File.**
(CSV)

**S2 File.**
(DOCX)

## Acknowledgments

We would like to think Dr. Virginie Callot for the acquisition of the images, Dr. Pierre-Jean Arnoux for his support and Wendy Silva-Verissimo for the discussion of the methodology and support along the study. This study was supported by the AIMCI fellowship.

## Author Contributions

**Conceptualization:** Patrice Sudres, Morgane Evin.

**Formal analysis:** Morgane Evin.

**Investigation:** Lugdivine Leblond, Morgane Evin.

**Methodology:** Lugdivine Leblond, Patrice Sudres, Morgane Evin.

**Supervision:** Morgane Evin.

**Validation:** Morgane Evin.

**Writing – original draft:** Lugdivine Leblond.

**Writing – review & editing:** Patrice Sudres, Morgane Evin.

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
