## [Editor Report · Decision Letter 0]

25 Sep 2023

PONE-D-23-26218Cerebro-spinal Flow Pattern in the Cervical Subarachnoid Space of Healthy Volunteers: Influence of the Spinal Cord morphologyPLOS ONE

Dear Dr. Evin,

Thank you for submitting your manuscript to PLOS ONE. After careful consideration, we feel that it has merit but does not fully meet PLOS ONE’s publication criteria as it currently stands. Therefore, we invite you to submit a revised version of the manuscript that addresses the points raised during the review process.

ACADEMIC EDITOR: Please review the translation of the ethical clearance provided with the submission. Since the study has incorporated humans as subjects, there has to be a study specific ethical approval which seems unclear on reviewing the submission. The authors are requested to revisit the submission files and address the issue before it can be sent to the reviewers for further processing.==============================

We look forward to receiving your revised manuscript.

Kind regards,

Sagar Panthi, MBBS

Academic Editor

PLOS ONE

A clean copy of the edited manuscript (uploaded as the new *manuscript* file)”.

 [Unfunded studies]. 

6. We notice that your supplementary figures are uploaded with the file type 'Figure'. Please amend the file type to 'Supporting Information'. Please ensure that each Supporting Information file has a legend listed in the manuscript after the references list.

7. We notice that your supplementary figures are included in the manuscript file. Please remove them and upload them with the file type 'Supporting Information'. Please ensure that each Supporting Information file has a legend listed in the manuscript after the references list.

---

## [Author Response · Author response to Decision Letter 0]

23 Oct 2023

Journal : PLOS-ONE

TITLE: 

Cerebro-spinal Flow Pattern in the Cervical Subarachnoid Space of Healthy Volunteers: Influence of the Spinal Cord morphology

NB: PONE-D-23-26218

OBJECTIVES: Responses to Reviewers’ comments 

C1 - Please review the translation of the ethical clearance provided with the submission. Since the study has incorporated humans as subjects, there has to be a study specific ethical approval which seems unclear on reviewing the submission. The authors are requested to revisit the submission files and address the issue before it can be sent to the reviewers for further processing.

Dear Editor, 

We are now provided a revised version of the translation of the ethical clearance. However, we would like to underline that the data have been previously published and that all the ethical approval were granted as we have been demonstrated with the provision of the ethical letter and its translation. Moreover, this study uses only 3D subarachnoidal canal segmentation to perform a computational fluid analysis within the subarachnoidal canal. As such and considering the fact that the initial data are already published and validated in term of ethics in [1].

1. Sudres P, Evin M, Arnoux P-J, Callot V. Cervical canal morphology: Effects of neck flexion in normal condition - New elements for biomechanical simulations and surgical management. Spine. 2020. doi:10.1097/BRS.0000000000003496

C2 - We note that Figure 1, 2B, 2D, 2E, 3B, 3D, 4, S1 in your submission contain copyrighted images. All PLOS content is published under the Creative Commons Attribution License (CC BY 4.0), which means that the manuscript, images, and Supporting Information files will be freely available online, and any third party is permitted to access, download, copy, distribute, and use these materials in any way, even commercially, with proper attribution. For more information, see our copyright guidelines: http://journals.plos.org/plosone/s/licenses-and-copyright.

As previously mentioned in exchange with the editor, all the figures were created by the authors and are not published elsewhere. As such, no copyright is requested. 

We sincerely hope that this new version of the translation will be suitable for you to accept transfer for review and would like to mentioned that considering the delay occasioned by this issue we would appreciate a quick reply regarding this manuscript. 

Best regards,

Morgane EVIN

---

## [Decision Letter · Decision Letter 1]

29 Jan 2024

PONE-D-23-26218R1Cerebro-spinal Flow Pattern in the Cervical Subarachnoid Space of Healthy Volunteers: Influence of the Spinal Cord morphologyPLOS ONE

Dear Dr. Evin,

Thank you for submitting your manuscript to PLOS ONE. After careful consideration, we feel that it has merit but does not fully meet PLOS ONE’s publication criteria as it currently stands. Therefore, we invite you to submit a revised version of the manuscript that addresses the points raised during the review process.

We look forward to receiving your revised manuscript.

Kind regards,

Sagar Panthi, MBBS

Academic Editor

PLOS ONE

Journal Requirements:

Reviewers' comments:

Reviewer's Responses to Questions

**Comments to the Author**

1. If the authors have adequately addressed your comments raised in a previous round of review and you feel that this manuscript is now acceptable for publication, you may indicate that here to bypass the “Comments to the Author” section, enter your conflict of interest statement in the “Confidential to Editor” section, and submit your "Accept" recommendation.

Reviewer #1: All comments have been addressed

Reviewer #2: (No Response)

2. Is the manuscript technically sound, and do the data support the conclusions?

Reviewer #1: Yes

Reviewer #2: Partly

3. Has the statistical analysis been performed appropriately and rigorously? 

Reviewer #1: Yes

Reviewer #2: Yes

4. Have the authors made all data underlying the findings in their manuscript fully available?

Reviewer #1: Yes

Reviewer #2: Yes

5. Is the manuscript presented in an intelligible fashion and written in standard English?

Reviewer #1: Yes

Reviewer #2: Yes

6. Review Comments to the Author

Reviewer #1: I thank the editor for providing me the chance to share my review insights on this professionally made study. though the study focuses on the craniocervical junction, I would like-however- to inquire regarding the wall shear stress and compression ratio in the 4th ventricular through the central canal as it has marked implications regarding the pathologies of CNS and spinal cord e.g. the specific predilection of Alcohol intoxication on midline inferior vermis.

Reviewer #2: Thank you for the opportunity to review this very interesting paper regarding the cerebro-spinal flow Pattern in the cervical subarachnoid space of healthy volunteers.

Here are my specific comments:

1. Abstract- too many abbreviations not commonly known- need to write them in full.

a. What is the significance of this findings in real world.

b. As the audience comprises of neurologist , neurosurgeons and other neuroscientists, the language used should be tailored to them in an understandable fashion.

2. Introduction – a bit long. Can be shortened to deliver the same meaning

3. Please add in the discussion section what could be the potential clinical application of this research.

7. PLOS authors have the option to publish the peer review history of their article (what does this mean?). If published, this will include your full peer review and any attached files.

Reviewer #1: **Yes: **Mohamed Mostafa

Reviewer #2: No

---

## [Author Response · Author response to Decision Letter 1]

7 Mar 2024

Journal : PLOS-ONE

TITLE: 

Cerebro-spinal Flow Pattern in the Cervical Subarachnoid Space of Healthy Volunteers: Influence of the Spinal Cord morphology

NB: PONE-D-23-26218

OBJECTIVES: Responses to Reviewers’ comments 

Reviewer #1: I thank the editor for providing me the chance to share my review insights on this professionally made study. though the study focuses on the craniocervical junction, I would like-however- to inquire regarding the wall shear stress and compression ratio in the 4th ventricular through the central canal as it has marked implications regarding the pathologies of CNS and spinal cord e.g. the specific predilection of Alcohol intoxication on midline inferior vermis.

We thank the reviewer for his/her comment and would like to underline that the study focuses on the cervical spine rather than cranio-cervical junction which will be furtherly study in the next work. Indeed, both the morphology and the fluid dynamics in the craniocervical junction need to be fully studied and was behind the scope of this study. Additionally, the study of the fluid dynamics between 4th ventricle and central canal could also be addressed using analytical model rather than computational fluid dynamic. The present work is a step forward more knowledge on the cerebro-spinal fluid dynamic in healthy cervical spine and brain before addressing pathological context such as Chiari. It is the first to describe correlation between morphology (compression ratio) and flow pattern. We truly hope that based on the presented results, this work will be accepted for publication. 

Reviewer #2: Thank you for the opportunity to review this very interesting paper regarding the cerebro-spinal flow Pattern in the cervical subarachnoid space of healthy volunteers.

We thank the reviewer for his/her helpful comment and hope to have addressed all and have improve the manuscript. 

Here are my specific comments:

1. Abstract- too many abbreviations not commonly known- need to write them in full.

All abbreviations have been removed from the abstract. 

a. What is the significance of this findings in real world.

 The abstract has been modified in order to highlight the need for cerebro-spinal flow quantification in clinical practice as well as the modification of the cerebro-spinal flow pattern depending on the morphology of the canal. 

b. As the audience comprises of neurologist, neurosurgeons and other neuroscientists, the language used should be tailored to them in an understandable fashion.

This has been modified through the abstract. 

2. Introduction – a bit long. Can be shortened to deliver the same meaning.

The introduction has been reduced and clarified. 

3. Please add in the discussion section what could be the potential clinical application of this research.

A final paragraph on potential clinical application as been added to the discussion.

---

## [Editor Report · Decision Letter 2]

23 Apr 2024

Cerebro-spinal Flow Pattern in the Cervical Subarachnoid Space of Healthy Volunteers: Influence of the Spinal Cord morphology

PONE-D-23-26218R2

Dear Dr. Evin,

We’re pleased to inform you that your manuscript has been judged scientifically suitable for publication and will be formally accepted for publication once it meets all outstanding technical requirements.

Kind regards,

Sagar Panthi, MBBS

Academic Editor

PLOS ONE

---

## [Editor Report · Acceptance letter]

29 Apr 2024

PONE-D-23-26218R2 

PLOS ONE

Dear Dr. Evin, 

I'm pleased to inform you that your manuscript has been deemed suitable for publication in PLOS ONE. Congratulations! Your manuscript is now being handed over to our production team.

Kind regards, 

on behalf of

Dr. Sagar Panthi 

Academic Editor

PLOS ONE